# A Novel Three-Gene Score as a Predictive Biomarker for Pathologically Complete Response after Neoadjuvant Chemotherapy in Triple-Negative Breast Cancer

**DOI:** 10.3390/cancers13102401

**Published:** 2021-05-16

**Authors:** Masanori Oshi, Fernando A. Angarita, Yoshihisa Tokumaru, Li Yan, Ryusei Matsuyama, Itaru Endo, Kazuaki Takabe

**Affiliations:** 1Department of Surgical Oncology, Roswell Park Cancer Institute, Elm & Carlton Streets, Buffalo, NY 14263, USA; masa1101oshi@gmail.com (M.O.); fernando.angaritacelis@roswellpark.org (F.A.A.); Yoshihisa.Tokumaru@roswellpark.org (Y.T.); 2Department of Gastroenterological Surgery, Yokohama City University Graduate School of Medicine, Yokohama, Kanagawa 236-0004, Japan; ryusei@yokohama-cu.ac.jp (R.M.); endoit@yokohama-cu.ac.jp (I.E.); 3Department of Surgical Oncology, Graduate School of Medicine, Gifu University, Gifu 501-1193, Japan; 4Department of Biostatistics & Bioinformatics, Roswell Park Cancer Institute, Elm & Carlton Streets, Buffalo, NY 14263, USA; li.yan@roswellpark.org; 5Department of Surgery, University at Buffalo, The State University of New York Jacobs School of Medicine and Biomedical Sciences, Buffalo, NY 14263, USA; 6Department of Breast Surgery and Oncology, Tokyo Medical University, Tokyo 113-8654, Japan; 7Department of Surgery, Niigata University Graduate School of Medical and Dental Sciences, Niigata 700-8558, Japan; 8Department of Breast Surgery, Fukushima Medical University, Fukushima 960-1295, Japan

**Keywords:** three gene, predictive biomarker, prognosis, neoadjuvant chemotherapy, triple-negative breast cancer, tumor immune microenvironment

## Abstract

**Simple Summary:**

Neoadjuvant chemotherapy is now a standard of care not only to decrease tumor size for breast conserving operation but also to assess drug response of an in situ cancer. Although the triple-negative subtype typically responds better compared to the other subtypes, a pathological complete response, which is a surrogate of survival, is achieved in less than half of the cases. For the most efficient patient selection, and avoiding unnecessary side effects and financial toxicity, an accurate predictive biomarker is urgently needed. We developed a novel three-gene score that associated with immune cell infiltration and pathological complete response not only in the training cohort but also in the validation triple-negative cohort. High-score TNBC was significantly associated with better survival in patients who received chemotherapy but not in patients who did not receive chemotherapy. Our score is a predictive and prognostic biomarker of response to neoadjuvant chemotherapy in triple-negative breast cancer patients.

**Abstract:**

Although triple-negative breast cancer (TNBC) typically responds better to neoadjuvant chemotherapy (NAC) compared to the other subtypes, a pathological complete response (pCR) is achieved in less than half of the cases. We established a novel three-gene score using genes based on the E2F target gene set that identified pCR after NAC, which showed robust performance in both training and validation cohorts (total of *n* = 3862 breast cancer patients). We found that the three-gene score was elevated in TNBC compared to the other subtypes. A high score was associated with Nottingham histological grade 3 in TNBC. Across multiple cohorts, high-score TNBC enriched not only E2F targets but also G2M checkpoint and mitotic spindle, which are all cell proliferation-related gene sets. High-score TNBC was associated with homologous recombination deficiency, high mutation load, and high infiltration of Th1, Th2, and gamma-delta T cells. However, the score did not correlate with drug sensitivity for paclitaxel, 5-fluorouracil, cyclophosphamide, and doxorubicin in TNBC human cell lines. High-score TNBC was significantly associated with a high rate of pCR not only in the training cohort but also in the validation cohorts. High-score TNBC was significantly associated with better survival in patients who received chemotherapy but not in patients who did not receive chemotherapy. The three-gene score is associated with a high mutation rate, immune cell infiltration, and predicts response to NAC in TNBC.

## 1. Introduction

Neoadjuvant chemotherapy (NAC) is commonly offered to patients with breast cancer, with the goals of downstaging the tumor to enable breast conservation and to assess in situ treatment response [1,2]. Triple-negative breast cancer (TNBC), defined by the absence of estrogen receptor (ER), progesterone receptor (PR), and HER2 amplification [3], is the most highly proliferative breast cancer subtyp [4]. Given that chemotherapy targets highly proliferative cells, TNBC is more likely to achieve a pathologic complete response (pCR) after NAC [5,6,7]. Consequently, pCR is typically considered a surrogate of survival. Indeed, chemotherapy improves the survival of patients with TNBC as noted across large clinical studies [8]. Unfortunately, less than 50% of patients with TNBC achieve pCR after NAC [6,9]. The remaining proportion of patients receive ineffective treatment with unnecessary side effects and delay of other cancer treatments. To this end, a biomarker that predict treatment response and survival would be beneficial to appropriately identify patients who will benefit from NAC, thereby reducing ineffective treatments and financial strain as well as improve patients’ quality of life.

The rapid advance in genomic technology as well as improvements in data sharing platforms have revolutionized the usage of gene expression data. Oncotype Dx and MammaPrint, which utilize expression data of a limited number of genes, are used in clinical practice to predict the benefit of adjuvant chemotherapy in hormone receptor-positive breast cancers. In addition, a number of algorithms have been developed to dissect the complex cancer biology within a human tumor [10,11,12,13]. Derived by integrating the expression of many related genes, these algorithms allow for a more accurate understanding of complex cancer biologies that are difficult to grasp with a single gene [14,15,16]. Because cytotoxic chemotherapeutics target proliferating cells, tumors with an active cell cycle are expected to respond to NAC. Our group previously reported that a G2M checkpoint and E2F target pathway score, both of which are genes essential to the cell cycle, predicted NAC response as well as prognosis in ER-positive/HER2-negative breast cancer but not in TNBC [17,18]. We also reported that breast cancer with a high mutation rate was associated with high cell proliferation but counterbalanced with anticancer immune cell infiltration [19]. Although using a scoring system based on 200 genes to define the complex tumor biology may be robust, we aimed to extract 3 genes from the previously published E2F target pathway score to develop an efficient tool. In this study, we hypothesized that our novel three-gene score is associated with cell proliferation, high immune cell infiltration, and predicts pCR after NAC in TNBC.

## 2. Results

### 2.1. Establishment of a Novel Three-Gene Score to Predict Pathological Complete Response (pCR) after Neoadjuvant Chemotherapy (NAC) in Triple-Negative Breast Cancer (TNBC)

We previously reported the E2F pathway score as a predictive biomarker of NAC response in ER-positive/Her2-negative breast cancer patients [17]. Given that TNBC responds better to NAC than the ER-positive/Her2-negative subtype, we hypothesized that there are several genes within the 200 genes of the E2F pathway score that may be particularly associated with pCR in TNBC. To identify those genes, we used gene expression profiles of TNBC in the GSE25066 cohort as the training cohort given its large sample size (*n* = 508). First, we compared the expression of 200 genes of the Hallmark E2F targets set between 57 TNBC patients who achieved pCR and 113 TNBC patients who did not in the GSE25066 cohort. With this differential gene expression analysis (DGEA), we found that the expression of nine genes was significantly associated with pCR (adjusted *p* < 0.05) (Figure 1A). We selected three genes with the highest statistical significance out of those nine (lowest *p* values), *CDKN2C*, *DEK*, and *MCM3*. In order to summarize the gene expression levels into a single measurement for further analysis, we weighted the individual gene expression levels using their DGEA log_2_(fold change [FC]) values and then summed the weighted values as the e-gene score in the following formula (1): 0.539914 × (expression^CDKN2C^) + 0.487451 × (expression^DEK^) + 0.313544 × (expression^MCM3^)(1)

The top one-third within each cohort was defined as a high score (Appendix A). In order to assess the predictive performance of the score, receiver operating characteristic-area under the curve (ROC-AUC) analysis was performed. The AUC of the score was 0.735, whereas that of the E2F pathway score was 0.628, suggesting that the 3-gene score had a superior predictive performance (Figure 1B; *p* = 0.015). The AUC of the 3-gene score was the highest compared to any other genes in the E2F targets gene set (Appendix A). The predictive efficiency of the 3-gene score was validated in two additional completely independent cohorts (Figure 1C; AUC = 0.742 in GSE20194 (TNBC; *n* = 68), and AUC = 0.747 in HESS (TNBC; *n* = 27)). These data suggested that the three-gene score had predictive properties to measure pCR after NAC in TNBC.

### 2.2. The three-Gene Score Was Highest in TNBC, and a High Score Was Associated with Advanced Nottingham Histological Grade in TNBC

Given that the three-gene score was generated from genes in E2F targets, which is one of the cell proliferation-related gene sets, we expected that the three-gene score was associated with the clinical aggressiveness of breast cancer. We found that the three-gene score was highest in TNBC among the subtypes in both the GSE25066 and METABRIC cohorts (Figure 2A; both *p* < 0.001). The three-gene score was significantly elevated in Her2 overexpressing, basal-like, and claudin-low subtypes in PAM50 classification of the METABRIC cohort (Appendix A). Within TNBC, Nottingham histological grade 3 was significantly associated with a high three-gene score compared to grade 1 and 2 (Figure 2B; both *p* < 0.001). The three-gene score was weakly correlated with *MKI67* gene expression in the GSE25066 and METABRIC cohort (Spearman rank correlation (*r*) = 0.410 [*p* < 0.01] and *r* = 0.208 [*p* < 0.01], respectively). The score was not associated with the American Joint Committee on Cancer (AJCC) pathological stage in either of the cohorts (*p* = 0.233 and 0.896, respectively).

### 2.3. A High Three-Gene Score TNBC Enriched Cell Proliferation-Related Gene Sets

Gene set enrichment analysis (GSEA) with the MSigDB hallmark gene set collection was performed to investigate the association of the three-gene score with the cancer biology of TNBC in three independent cohorts (GSE25066, METABRIC, and TCGA). The top one-third was defined as a high score. A high three-gene score TNBC significantly enriched cell proliferation-related gene sets (E2F targets, G2M checkpoint, and mitotic spindle) consistently in all three cohorts (Figure 3). These results suggested that the three-gene score reflects not only E2F targets gene sets, but also cell proliferation, which is in agreement with the notion that highly proliferative cancer responds to cytotoxic chemotherapy better than the ones that do not.

### 2.4. A High Three-Gene Score TNBC Is Associated with Homologous Recombination Deficiency (HRD), High Mutation Rate, and Have High Infiltration of Gamma-Delta (γδ) T Cells, T Helper Type 1 Cells, and T Helper Type 2 Cells

We have previously reported that some highly proliferative breast cancers are associated with HRD and high mutation load [19]. Therefore, it was of interest to investigate whether a high three-gene score TNBC was related with HRD and mutation load. Using the calculated scores on the TCGA cohort by Thorsson et al. [20], we found that high three-gene score TNBC was significantly associated with high HRD, silent and non-silent mutation load, amount of fraction altered, and single-nucleotide variant (SNV) neoantigens (Figure 4A; *p* < 0.001, *p* = 0.007, *p* = 0.003, *p* < 0.001, and *p* = 0.011, respectively). The three-gene score was not associated with *BRCA1* and *BRCA2* mutations, which are known DNA repair genes, as well as *PD-L1* expression in TNBC of the METABRIC cohort (Appendix A).

It is well known that tumor-infiltrating lymphocytes (TILs) play a critical role in treatment response and prognosis in breast cancer [21]. High TIL infiltration also results in better response to NAC [22]. Together with the fact that high three-gene score TNBC is associated with high mutation load, it was of interest to study the association of the three-gene score with infiltrating immune cells in TNBC. Using the xCell algorithm, we examined the association of the score with a fraction of immune cells in TNBC of the GSE25066 and METABRIC cohorts. A high three-gene score was significantly associated with high fraction of T helper type1 (Th1) and type2 (Th2) cells and gamma-delta (γδ) T cells consistently in both cohorts (Figure 4B; Th1; *p* = 0.019 and 0.013, Th2; both *p* < 0.001, γδT; both *p* < 0.001, in the GSE25066 and METABRIC, respectively). A high three-gene score was associated with low fraction of regulatory T cells (Tregs) and M2 macrophages in the GSE25066 cohort but not in the METABRIC cohort (Figure 4C). Additionally, we found that the three-gene score was elevated not only in cancer cells but also in immune cells as well in the single-cell sequence breast cancer cohort (GSE75688, Figure 4D). These findings suggest that a high three-gene score is associated with high mutation load and infiltration of Th1, Th2, and γδT, which is in agreement with the previously proposed notion in TNBC.

### 2.5. The Three-Gene Score Did Not Correlate with Sensitivity to Chemotherapy in TNBC Cell Lines

Given that the three-gene score was elevated in both immune cells and cancer cells in the human tumor microenvironment, it was of interest whether the score was associated with drug sensitivity in breast cancer cells. We investigated the association of the three-gene score with drug sensitivity using a breast cell line cohort (CCLE) that did not include immune cells. We found that the level of three-gene score expression was not correlated with the level of area under the curve (AUC) for paclitaxel, 5-fluorouracil, cyclophosphamide, and doxorubicin in TNBC cell lines (details of the cell lines are shown in Appendix A) (Figure 5; *r* = 0.069, 0.122, −0.110, and −0.040, respectively, all *p* > 0.5). These data suggest that the in vitro results of the three-gene score expression may not be directly translatable to the clinical setting.

### 2.6. A High 3-Gene Score Was Associated with a Significantly Improved pCR Rate after NAC in TNBC Patients but Not in Cell Lines

Based on our findings that the three-gene score is associated with mutation load, immune cell infiltration, and cell proliferation, which are all biological features of better response to NAC, we expected that a high three-gene score is predictive of pCR in any TNBC cohort. We found that TNBC with a high three-gene score prior to the treatment was associated with a significantly higher pCR rate not only in the training GSE25066 TNBC cohort (taxane and anthracycline) but also in the other two validation cohorts, GSE20194 (paclitaxel, 5-fluorouracil, doxorubicin, and cyclophosphamide) and HESS (paclitaxel, fluorouracil, doxorubicin,, and cyclophosphamide), which have less patient numbers (Figure 6; *p* < 0.001, *p* = 0.003, and *p* = 0.046, respectively). These results suggest that the three-gene score is a predictive biomarker of pCR after NAC in TNBC patients.

### 2.7. A High Three-Gene Score Was Significantly Associated with Better Survival in Patients with TNBC Who Underwent Chemotherapy

pCR after NAC is used as a surrogate to predict survival in breast cancer patients [1,2]; however, it is not uncommon for a biomarker that predicts pCR to not be associated with improved survival [17,18]. To this end, it was of interest to determine whether a high three-gene score TNBC was associated with better survival. As expected, the high three-gene score TNBC group was significantly associated with better disease-free survival (DFS) in the training GSE25066 cohort (Figure 7A; *p* = 0.031). Interestingly, this was also the case in the patients who received chemotherapy (Figure 7B; overall survival [OS]; *p* = 0.011, DFS; *p* = 0.049, and disease-specific survival [DSS]; *p* = 0.035) but not in patients who did not receive chemotherapy (OS; *p* = 0.747, DFS; *p* = 0.685, and DSS; *p* = 0.890) in the validation METABRIC cohort. These findings suggest that the three-gene score was associated with not only a better response to NAC but also with chemotherapy in general that prolongs survival in TNBC.

## 3. Discussion

We established a novel three-gene score using genes derived from a previously published E2F target gene set that predicts pCR after NAC. We found that the three-gene score was elevated in TNBC compared to other subtypes. A high score was associated with Nottingham histological grade 3 in TNBC, which indicated enhanced cancer cell proliferation. High three-gene score TNBC enriched not only E2F targets but also G2M checkpoint and mitotic spindle, which are all cell proliferation-related gene sets, in training and two other large validation cohorts. A high three-gene score was associated with HRD and high mutation load, as well as with high infiltration of Th1, Th2, and gamma-delta T cells. Surprisingly, the three-gene score did not correlate with drug sensitivity for paclitaxel, 5-fluorouracil, cyclophosphamide, and doxorubicin in TNBC human cell lines. However, the score was significantly associated with pCR not only in the training cohort, which underwent taxane and anthracycline, but also in the validation cohorts, which underwent paclitaxel, 5-fluorouracil, doxorubicin, and cyclophosphamide. Interestingly, the high three-gene score was consistently associated with overall, disease-free, and disease-specific survival of the patient who underwent chemotherapy but not with the survival of patients who were not in the METABRIC cohort. This result suggests that the three-gene score is a predictive biomarker rather than a prognostic biomarker; however, the score was associated with disease-free survival in the GSE25066 cohort, which had treatment data. In order to confirm the practical utility of the three-gene score, a prospective study with detailed use of chemotherapy agents is needed.

Anthracyclines and taxanes are the most commonly used chemotherapies that have survival benefit for breast cancer [8]. It is well known that TNBC is more likely to achieve pCR after NAC compared to the other subtypes [5,6,23,24]; however, this is less than half of the patients who undergo NAC. As an example, the first-line remission rate was 36% in the UNICANCER-PACS 05 trial [25]. This is with the expense of major adverse events, including hematological toxicity, alopecia, and cardiotoxicity for anthracyclines and persistent neuropathy for taxanes. Therefore, a predictive biomarker of NAC will practically help to achieve appropriate patient selection to maximize the benefit and minimize the risk of side effects, and thus, it is urgent needed. We expect our three-gene score to be practically useful for patient selection for NAC in TNBC patients.

There are several accepted parameters to estimate the probability of pCR. Clinical phenotypes, such as subtype, grade, age, and hormone receptor status, after two courses of chemotherapy have been reported [24,26]. Several trials have shown that the lack of response after the first two cycles predicts that pCR is unlikely even after completion of chemotherapy [27,28]. Following the positive results of the prospective randomized clinical trials, TAILORx [29] and MINDACT [30], gene expression panels OncotypeDX and MammaPrint are now standard of care to predict the effect of chemotherapy. Masuda et al. [31] reported a promising correlation between the seven subtypes of TNBC and response to NAC. To this end, it is of interest to compare the predictive value of the seven subtypes of TNBC and the three-gene score; however, we were unable to do so given that we do not have data on the subtype distribution of TNBC in the study population.

In this study, we established a novel predictive biomarker of NAC response in TNBC patients using the expression of three genes based on the E2F targets gene set in tumors. This is based on the fact that cytotoxic chemotherapy acts on the proliferating cells in which the E2F pathway is activated. We chose the three genes with the highest statistical significance with pCR after NAC given that minimizing the number of constituent genes improves the clinical utility of the score. Cyclin-dependent kinase 4 inhibitor C (*CDKN2C*), which is known as p18^INK4C^, is a member of the *INKC* family, which inhibits *CDK4* or *CDK6* and regulates the cell cycle in thee G1 phase. Currently, inhibitors targeting *CDK4/6* activity (abemaciclib, palbociclib, and ribociclib) have been approved for clinical use in breast cancer patients [32,33]. *DEK* is known as an oncogene and is overexpressed in multiple cancers, such as melanoma [34], gastric cancer [35], and breast cancer [36]. Overexpression of *DEK* is associated with cancer cell proliferation and migration as well as chemoresistance [36,37]. Recently, it has been shown that *DEK* induces M2 macrophage polarization and creates an immune-suppressed tumor microenvironment [38]. Mini-chromosome maintenance 3 (*MCM3*), a member of the *MCM* family, is associated with DNA replication [39]. MCM protein increases gene expression and interaction with retinoblastoma protein and regulates cell proliferation. The *MCM3* gene was shown to promote cell replication and reflect cancer cell proliferation in breast cancer [40]. *MCM5* is the same family as *MCM3*, and the AUC and *p*-value of *MCM5* was almost the same as *MCM3*. However, when *MCM5* was used in the three-gene score instead of *MCM3*, a high three-gene score did not significantly associate with better survival in the chemotherapy (+) group in the METABRIC cohort, as shown in Appendix A. Because there was no strong correlation between the expression level of *MCM3* and *MCM5* (Appendix A), we speculate that the effect of the expression levels on each gene on clinical outcome may be different, even within the same family. We used coefficients of each gene to establish the score. The coefficients are used to weigh the biomarker value of each of the genes so that a gene with a higher value has more importance in the formula that is used to reduce the three-gene expression measurements into a single value. It was somewhat unexpected that the three-gene score correlated weakly with *MKI67* expression, which is the most commonly used marker of cell proliferation in clinical practice. However, the score did strongly enrich the E2F targets, G2M checkpoint, and mitotic spindle gene sets, and was significantly associated with advanced histological grade. Given these results, we speculate that the three-gene score may reflect cancer cell proliferation that is less involved with *MKI67* production.

High infiltration of CD8^+^ T cells is known to correlate with pCR after NAC [21,41]. On the other hand, these are not necessary and several immune cells other than CD8^+^ T cells have also been reported to be associated with pCR after NAC [42]. Our group previously reported that high infiltration of CD8^+^ T cells was significantly associated with better survival but not with pCR after NAC in TNBC [43]. Further, we reported that infiltration of regulatory T cells was significantly associated with pCR [44]. With this said, we believe that it may be inappropriate to use the three-gene level as a representation of specific fraction of infiltrating immune cells in the tumor microenvironment, because it is not made with its surface marker, and thus multiple types of cells can express those genes. Indeed, we demonstrated that the three-gene score can be high in multiple types of cells by the single-cell sequence cohort. Further investigation of the relationship between the three-gene score and immune cells within breast cancers is warranted.

The response to NAC is a short-term outcome, but it has been reported that it is also associated with long-term outcomes and has a significant impact on the patient’s survival. Therefore, pCR post-NAC is used as a prognostic marker in breast cancer. Indeed, patients with pCR after NAC were shown to have better survival compared with non-pCR patients in multiple large-scale clinical trials [6,45], and Cortazar et al. reported that pCR is a surrogate of improved survival by the CTNeoBC pooled analysis [46]. On the other hand, some question the association between pCR and patient survival. Although with less than 4 years of follow up, Tan et al. reported that there was no significant difference in survival between patients who did or did not achieve pCR after NAC in 518 breast cancer patients [26], and similar results have been reported by other groups [47,48]. Our group also previously reported that several factors that associated with pCR after NAC do not associate with survival in breast cancer patients [17,18,44]. In the current study, the three-gene score was significantly associated not only with pCR after NAC, but also with survival in TNBC patients, which was consistent in multiple independent cohorts. Although the three-gene score has been shown to be related to clinical outcomes in clinical samples, the three-gene score did not correlate with sensitivity of chemotherapy in vitro. Both in vitro and in vivo models are essential tools to elucidate cancer biology, whereas it is difficult to reproduce the complex human tumor environment using them. Given that the expression of the three-gene score by immune cells and tumor cells was comparable, assessing drug responses by ignoring the presence of immune cells may hide accurate information. We cannot help but speculate that our three-gene score can be both a predictive and prognostic biomarker for TNBC.

Although this study shows that the three-gene score can be a predictive and prognostic biomarker for TNBC, there are some limitations. First, the retrospective nature of our study prevents a robust conclusion on the definitive predictive role of the score. However, we included multiple independent cohorts, both for training and validation cohorts. In the future, a prospective study is needed to conclude that the three-gene score is a clinically useful predictive biomarker in breast cancer management. Another limitation is that there are few cohorts that contain details on relevant clinical information; therefore, we were unable to conduct further analyses, such as formulation of a nomogram. This is a known limitation of using datasets from publicly available databases; however, analyses of the cohort of one’s institution is cost prohibitive, and the sample size is limited.

## 4. Materials and Methods

### 4.1. Breast Cancer Cohorts and Their Data

To obtain the clinical and transcriptome data of breast cancer, the Gene Expression Omnibus (GEO) repository was utilized to access the studies of Symmans et al. (GSE25066; *n* = 508, regimen; taxane and anthracycline) [49] and Shi et al. (GSE20194; *n* = 248, regimens; paclitaxel, 5-fluorouracil, cyclophosphamide and doxorubicin) [50]. University of California Santa Cruz (UCSC) Xena was used to access data from Hess et al. (*n* = 133, regimens; paclitaxel, fluorouracil, doxorubicin, and cyclophosphamide) [51]. cBioPortal [52] was used to access The Cancer Genome Atlas (TCGA) Pan-Cancer study (TCGA PanCancer Atlas; *n* = 1069) [53], which selected female breast cancer patients, and Molecular Taxonomy of Breast Cancer International Consortium (METABRIC) study (*n* = 1904) [54], as we previously reported [43]. For the genes with multiple probes, the average value was used. The silent and non-silent mutation rate, fraction altered, single nucleotide variant (SNV) and indel neoantigens, and intratumor heterogeneity scores were obtained from a study by Thorsson et al. [20] in the TCGA cohort. The log_2_-transform of gene expression data was used in all analyses.

### 4.2. Gene Set Expression Analyses

Gene set enrichment analysis (GSEA) [55] with hallmark gene sets of the Molecular Signatures Database [56] was performed to explore the signaling pathways related to high and low 3-gene score expression in breast cancer, as we previously reported [57,58,59].

### 4.3. Statistical Analysis

R software (version 4.0.1) was used for statistical analyses. Group comparisons were performed using the Kruskal–Wallis test, Mann–Whitney U test, or Fisher’s exact test accordingly. The survival plot was plotted by the Kaplan–Meier method with the log-rank test. Statistical significance was set at a *p*-value < 0.05.

## 5. Conclusions

We established a novel three-gene score that was associated with cell proliferation, mutation, and infiltration of anticancer immune cells, and can be a predictive and prognostic biomarker of NAC response in patients with TNBC.

## Figures and Tables

**Figure 1 cancers-13-02401-f001:**
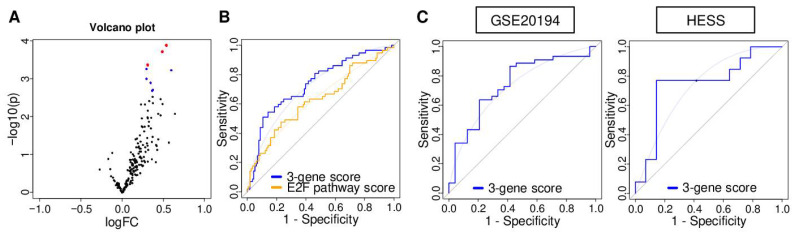
Establishment and association of the 3-gene score with response to neoadjuvant chemotherapy (NAC) for triple-negative breast cancer (TNBC). (**A**) Volcano plots illustrating the differentially expressed mRNAs between pathological complete response (pCR) (*n* = 57) and non pCR groups (*n* = 113) of TNBC in the GSE25066 cohort. *X*-axes; log_2_ (fold change), *Y*-axes; −log_10_ *p*-value from limma analysis. mRNA with adjusted *p*-value < 0.05 are marked in blue, and top three genes of *p*-value are marked in red. (**B**) Receiver operating characteristic (ROC) curve of the 3-gene score and E2F targets score with the area under the curve (AUC) in the GSE25066 cohort. (**C**) ROC curve of the 3-gene score with AUC in the GSE20194 and HESS cohorts.

**Figure 2 cancers-13-02401-f002:**
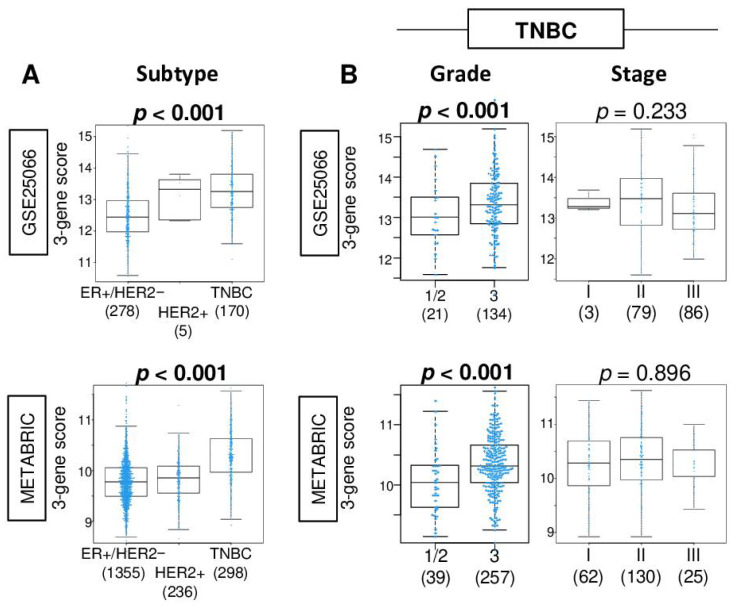
Association of the 3-gene score level with clinical characteristics in the GSE25066 and METABRIC cohorts. Boxplots of the 3-gene score level by (**A**) subtype in the whole cohort, and (**B**) Nottingham pathological grade (grade 1 and 2 vs. grade 3) and American Joint Committee on Cancer pathological stages in TNBC. Correlation plots between the 3-gene score and *MKI67* gene expression. The Kruskal–Wallis test or Mann–Whitney U test was used accordingly.

**Figure 3 cancers-13-02401-f003:**
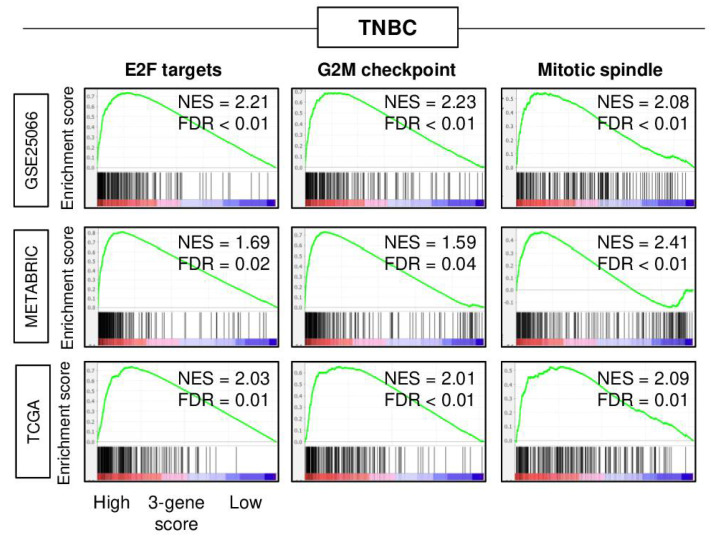
Gene set enrichment analysis (GSEA) of high 3-gene score triple-negative breast cancer (TNBC) in the GSE25066, METABRIC, and TCGA cohorts. Enrichment plots of hallmark E2F targets, G2M checkpoints, and mitotic spindle gene sets in the GSE25066, METABRIC, and TCGA cohorts superimposed with a normalized enrichment score (NES) and false discovery rate (FDR) are shown. NES and FDR were determined with the classical GSEA method, where FDR < 0.25 is considered significant.

**Figure 4 cancers-13-02401-f004:**
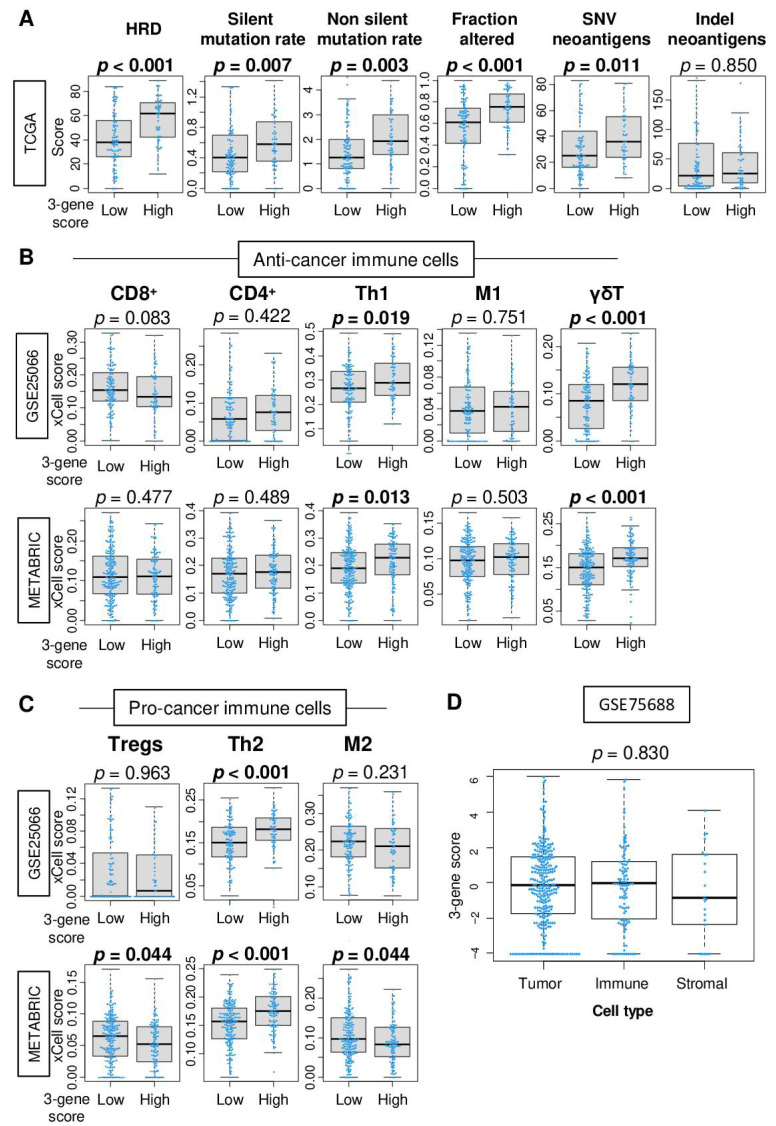
Association of the 3-gene score with mutation load and tumor-infiltrating immune cells. (**A**) Boxplots of the level of the mutation-related score; homologous recombination deficiency (HRD), silent and non-silent mutation load, fraction altered, single-nucleotide variant (SNV), and indel neoantigens by low and high 3-gene score triple-negative breast cancer in the TCGA cohorts. Boxplots of the fraction of (**B**) anti-cancer immune cells; CD8^+^ T cells, CD4^+^ T cells, T helper type 1 (Th1) cells, M1 macrophages, γδT cells and (**C**) pro-cancer immune cells; Regulatory T cells (Tregs), T helper 2 (Th2) cells, M2 macrophages, by low and high 3-gene scores TNBC in the GSE25066 and METABRIC cohorts. (**D**) Boxplots of the 3-gene score by tumor, immune, and stromal cells in single-cell sequence data (GSE75688 cohort). Mann–Whitney U or Kruskal–Wallis test were used to calculate *p* values.

**Figure 5 cancers-13-02401-f005:**
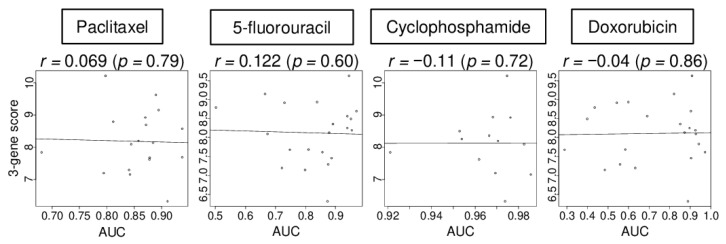
Correlation of the 3-gene score with treatment response in cell lines. Correlation plots between the 3-gene score level and area under the curve (AUC) of several drug sensitivity, paclitaxel, 5-fluorouracil, cyclophosphamide, and doxorubicin, for TNBC cell lines. Spearman rank correlation was used for the analysis.

**Figure 6 cancers-13-02401-f006:**
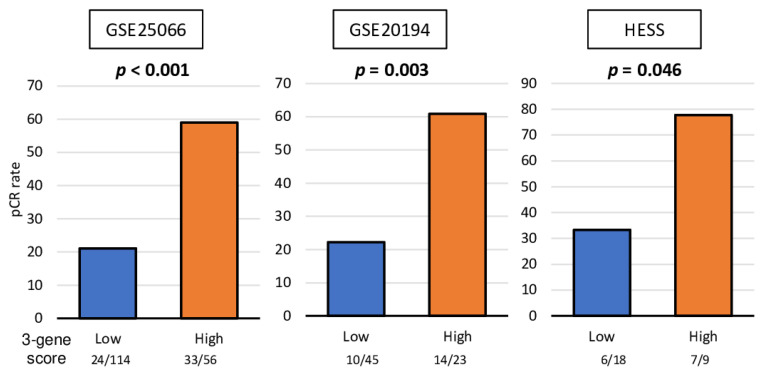
Association of the 3-gene score with drug response for TNBC cell lines and patients. Bar plots of the comparison of the pCR rate after NAC between the 3-gene score low (blue) and high (orange) groups in the GSE25066 (*n* = 170), GSE20194 (*n* = 68), and HESS (*n* = 27) cohorts. Fisher’s exact test was used for the analysis. Group sizes are shown underneath the bar.

**Figure 7 cancers-13-02401-f007:**
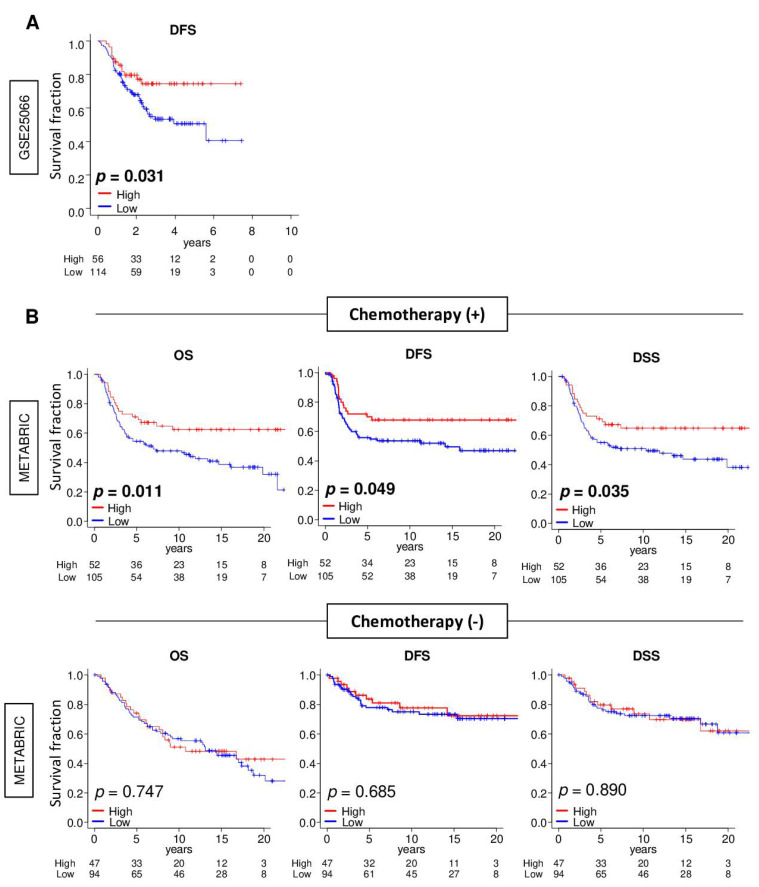
Association between the 3-gene score and the survival of patients with triple-negative breast cancer (TNBC) with or without chemotherapy. (**A**) Kaplan–Meier plots of comparison between low (blue line) and high (red line) 3-gene score groups for disease-free survival (DFS) in the GSE25066 cohort. (**B**) Kaplan–Meier plots of the comparison between low and high 3-gene score groups for overall survival (OS), DFS, and disease-specific survival (DSS) in the treatment group and non-treatment group in the METABRIC cohort. The top one-third was defined as the high-score group within the cohort. The log rank test was used to calculate the *p* values.

## Data Availability

All data were from previous studies.

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
