# Peer review of "A Novel Three-Gene Score as a Predictive Biomarker for Pathologically Complete Response after Neoadjuvant Chemotherapy in Triple-Negative Breast Cancer"

_cancers, 2021, doi:10.3390/cancers13102401_

Round 1
Reviewer 1 Report
The authors addressed all my concerns.
Reviewer 2 Report
The questions and comments addressed appear to have been answered appropriately.
Reviewer 3 Report
The authors fully addressed the comments I provided. The paper improved in clarity and in scientific soundness and can be considerted for pubblication in its present form
This manuscript is a resubmission of an earlier submission. The following is a list of the peer review reports and author responses from that submission.
Round 1
Reviewer 1 Report
In the present paper Masanori Oshi et al. establishes a novel predictive molecular biomarker of NAC response in TNBC patients, using the expression of three genes (CDKN2C, DEK, MCM3), based on EF2 targets gene set in tumor.
The authors show the 3-gene score is elevated in TNBC compare to other subtypes and is associated with some relevant biological characteristics which describe a TNBC aggressive behaviour, including: grade 3 enrichment, high expression of cell-proliferation genes, HRD and high mutation load as well as high TILs. Eventually, the 3-gene score is significantly associated with pCR after NAC, as observed both in the training cohort and in two independent validation cohorts of TNBC patients, retrospectively evaluated.
This scientific effort mirrors several other reports that investigate prognostic/predictive biomarkers (clinicopathological and/or molecular) to optimize the TNBC treatment strategy, with the identification of patients who will benefit or not from NAC, adding valuable information possibly relevant in clinic.
MAJOR COMMENTS
- The major limitation of the study is the retrospective nature of the analysis, as stated by the authors in the Such an analysis prevents any robust conclusion about the definitive predictive role of the candidate 3-gene score biomarker and the study should be considered as hypothesis generating, while is supposed the need for clinical validation in a next possible prospective study. An authors’ comment on that would be more than appreciated.
- Another limitation is the adoption of datasets from publicly databases. This research strategy could prevent the removal of heterogeneities among different patients’ populations and could lack some relevant clinical information. Since, it is well known how is crucial to systematically integrate the molecular data and clinicopathological characteristics for a broad application of any predictive tool (J. Sparano NEJM 2019), I would suggest the authors to consider this kind of clinical and molecular integration, at least in a descriptive, synoptic, multivariate analysis. Actually, with the available clinicopathological information (TNBC subtypes, clinical stage, type of chemo), the authors could also consider the development of a nomogram for the prediction of pCR, that integrates clinicopathological characteristics and the 3-gene score data.
MINOR COMMENTS
- Introduction (pg 2 line 67). OncotypeDX and MammaPrint gained LoE 1 for prediction because of 2 prospective randomized clinical trials testing the marker (TAILORx and MINDACT)
- Previous studies have managed to investigate the impact of the heterogeneous subtypes of TNBC in term of chemo-sensitivity. Following the 7 subtypes of TNBC, Masuda et al. (CCR 2013) confirmed the promising correlation b/w the cancer heterogeneity and response to NAC in TNBC. Could the authors report the subtypes distribution of TNBC in the study population ?
- Results (pg 3 line 100). Is there any cut-off in the 3-gene biomarker to define high low score ?
- Results (pg 3 line 110). Please report the crude number of TNBC patients included in the 2 independent validation cohorts of GSE20194 and HESS.
- Results (pg 4 line 144). The significant association b/w the 3-gene score and the cell proliferation targets genes are not in-line with the weak association observed b/w the 3-gene score with MKi67 gene expression. Please comment on that
- Results (pg 5 line 14). Along with the association with HRD/mutation load as well as with TILS, could the authors report the possible association of the 3-gene score with BRCA mutation and PDL1, respectively?
- Results (pg 8 line 230 and line 308). The OS advantage, only observed in case of high 3-gene score while on chemotherapy, suggests the lack of any prognostic role of the biomarker, supporting its predictive role. Please comment on that.
- Discussion (pg 9 line 291). The surrogacy value of the pCR at patient-level is well described in the CTNeoBCmetanalysis of P. Cortazar (Lancet 2014). Please mention and comment on that.
Reviewer 2 Report
The authors have reported that the novel three-gene score that they developed was useful ad predictive marker for pathological complete response (pCR) to neoadjuvant chemotherapy (NAC) in triple negative breast cancer (TNBC). The contents of the study is timely and are of general interest for breast cancer researchers. However, several points are unclear in the present form and could be improved.
1. In Figure 7B, From OS in METABRIC data, it appears that in the high 3-gene score group the chemotherapy (+) subgroup showed improvement in survival compared with the chemotherapy (-) subgroup. On the other hand, from DFS and DSS data, it appears that in high 3-gene score group the prognosis did not differ between chemotherapy (+) and chemotherapy (-) subgroups. Instead, in the low 3-gene score group, the prognosis of chemotherapy (+) subgroup appears to be worse than that of chemotherapy (-) group. They should make sure these possibilities from these data, and interpret the results appropriately in Discussion. From these results, the authors’ statement that “high 3 gene score was significantly associated with better survival in patient who received chemotherapy compared with who did not” does not appear very accurate.
2. In Result 2.2 and Figure 2, they describe about the relationship between three-gene score and AJCC stage. In this case, they should clarify if they used AJCC anatomic stage or prognostic stage.
3. To the knowledge of this reviewer, high tumor infiltrating CD8+ T cells are correlated with pCR and better prognosis after chemotherapy in TNBC. However, in this study, 3-gene score was not correlated with CD8+ T cells but correlated with Th1 and Th2 cells, but, nonetheless, the score was not correlated with CD4+ T cells. These apparently inconsistent phenomena should be discussed more in detail.
4. In Results 2.1, the authors present a formula to calculate 3-gene score. The rationale of coefficient for expressions of CDKN2C, DEK, and MCM3 should be explained more in detail.
5. In Figure 2, the number of cases of each group in Subtype, TNBC (Grade, Stage) should be described somewhere.
Reviewer 3 Report
This manuscript by Oshi et al., presented a novel scoring with 3 genes in TNBC.
They showed that the patients with the high score of 3 genes are a significantly good prognosis.
Their markers will helpful to the diagnosis of TNBC patients. It's good analysis strategies in the whole paper.
However, there are several concerns for the acceptance in the Cancers.
1. The authors used TNBC classification, but TNBC patients can be more divided into 'Basal-like' and 'claudin-low' in METABRIC cohorts.
When using the classification 'Basal-like' and 'claudin-low', are there differences in the 3 gene score?
2. The authors choose 3 genes from the results of the reanalysis. In figure 1A, the volcano plot showed many differential expressional genes. The authors should describe the reason to choose '3' number of genes.
3. In Lines 107-108 of Page 3, The author said 'The AUC of the 3-gene score was the highest compared to any other genes in the E2F target gene sets(Table S1)'. But Table S1 showed the there are MCM5 and TRA2B with high AUC score.
MCM5 gene is the same family of MCM3 which is chosen as 3 gene score. How about the usefulness of MCM5 as a marker for the aim of present study?
4. Lines 101-102 of Page 3, the authors show the formula of 3 gene score. The authors used some coefficients for the calculation, the authors must explain the reason of using the coefficients.
5. As Fig4D, the boxplots in the whole paper should be shown with the score of individual samples as dots. It is not required, but I strongly recommend it.
6. I recommend the reduction of self-citation. As expected,there are many self-citation, so I recommend to focus on only the important ones.
